# Evaluating Changes in Trauma Epidemiology during the COVID-19 Lockdown: Insights and Implications for Public Health and Disaster Preparedness

**DOI:** 10.3390/healthcare11172436

**Published:** 2023-08-31

**Authors:** Mariusz Jojczuk, Jakub Pawlikowski, Piotr Kamiński, Dariusz Głuchowski, Katarzyna Naylor, Jakub Gajewski, Robert Karpiński, Przemysław Krakowski, Józef Jonak, Adam Nogalski, Dariusz Czerwiński

**Affiliations:** 1Department of Trauma Surgery and Emergency Medicine, Medical University of Lublin, 20-081 Lublin, Poland; mariuszjojczuk@umlub.pl (M.J.); piotr.kaminiski@umlub.pl (P.K.); adam.nogalski@umlub.pl (A.N.); 2Department of Humanities and Social Medicine, Medical University of Lublin, 20-059 Lublin, Poland; jakub.pawlikowski@umlub.pl; 3Department of Computer Science, Faculty of Electrical Engineering and Computer Science, University of Technology, 20-618 Lublin, Poland; d.gluchowski@pollub.pl (D.G.);; 4Independent Unit of Emergency Medical Services and Specialist Emergency, Medical University of Lublin, Chodzki 7, 20-093 Lublin, Poland; 5Department of Machine Design and Mechatronics, Faculty of Mechanical Engineering, University of Technology, 20-618 Lublin, Poland; j.gajewski@pollub.pl (J.G.);; 6I Department of Psychiatry, Psychotherapy and Early Intervention, Medical University of Lublin, 20-439 Lublin, Poland; 7Orthopaedic and Sports Traumatology Department, Carolina Medical Center, Pory 78, 02-757 Warsaw, Poland

**Keywords:** trauma, COVID-19, SARS-CoV-2, epidemiology, injury patterns, readiness, public health, hospital admission trends

## Abstract

The COVID-19 pandemic demanded changes in healthcare systems worldwide. The lockdown brought about difficulties in healthcare access. However, trauma still required further attention considering its modifications. The presented study aims to investigate the variances in epidemiological patterns of trauma during the lockdown and the previous year, with a view to better understand the modifications in healthcare provision. The authors analyzed data from the first lockdown in 2020 (12 March–30 May) and the same period in 2019 from 35 hospitals in Lublin Province. A total of 10,806 patients in 2019 and 5212 patients in 2020 were included in the research. The uncovered changes adhered to the total admissions and mortality rate, the frequency of injuries in particular body regions, and injury mechanisms. The lockdown period resulted in a reduction in trauma, requiring an altered approach to healthcare provision. Our research indicates that the altered approach facilitated during such periods is essential for delivering tailored help to trauma patients.

## 1. Introduction

Injuries are a global health issue and a leading cause of death worldwide. According to the World Health Organization (WHO), 75 million people sustain an injury and more than 5 million die per year [1]. As reported by the Centers for Disease Control and Prevention (CDC), 246,041 trauma victims died, accounting for 6.1% of total deaths, in 2019 [2]. Trauma care has improved in recent decades, and injury prevention programs have become a major topic in public health, but the incidence and mortality of injuries remain high [3].

At the end of 2019, a new type of coronavirus was identified in Wuhan, China, and soon enough, the global pandemic of COVID-19 affected societies all over the world [4]. Particularly, in the initial months, many difficulties were observed. Social distancing, lockdown policies, and rearranging hospital wards in order to provide care for infected patients influenced the healthcare services provided to non-infected patients. In light of the massive increase in demand for acute care beds and manpower, the postponement of non-urgent procedures and the relocation of resources were ordered in many countries. Previous studies reported a change in trauma epidemiology influenced by the imposed restrictions. A significant decrease in trauma admissions during the COVID-19 lockdown was found [5,6,7,8]. Differences in the most common injury mechanism and mean age were observed during COVID-19 compared to the same period in previous years [9,10,11]. A broad review by Waseem et al. about the impact of lockdown on the presentation and management of trauma showed an overall reduction in trauma referrals with an increase in trauma occurring at home, worldwide [12]. A recent systemic review by Antonini et al. demonstrated a significant decrease in major trauma admissions, although without changes in the severity of injury [13]. A pandemic of this magnitude is rare and provides a unique opportunity to evaluate trends in trauma. The obtained results may help to discover areas of improvement in future pandemics and allow for the appropriate management of resources in the future.

The first case of COVID-19 infection in Poland was identified in March 2020 [14]. In light of the experience gathered from other European countries, where the early stage of the pandemic disrupted healthcare systems, the Polish government decided to introduce a first national lockdown on 14 March 2020 [15]. The most severe restrictions were applied between 25 March 2020 and 20 April 2020 [16]. In the above-mentioned period, public gatherings and non-essential movements outside of places of residence were banned. Shopping malls, sports centers, schools, restaurants, and cultural institutions were closed. Subsequently, after 20 April 2020, restrictions were gradually lifted until 30 May 2020, when they were cancelled. In Poland, compared to the other European countries, the COVID-19 restrictions were implemented relatively early which resulted in the limited spread of the virus. According to the Polish Ministry of Health, 23,571 confirmed COVID-19 cases and 1061 deaths were reported on 30 May 2020 [17]. In the first lockdown period, the absolute number of cases and deaths in Poland was lower compared to western European countries [18]. Also, numerous hospitals were converted into so-called one-named hospitals (19 health facilities specialized in COVID-19), creating another barrier to medical care provision for trauma patients [19].

The use of numerical, statistical, and machine-learning methods can complement typical experimental studies and may allow a better understanding of the complex relationships commonly found in medicine [20,21,22,23]. Statistical analyses based on the ICD-10 classification are particularly useful in trauma research because they allow the classification and comparison of different types of injuries and their prevalence in the population [24]. This allows doctors and researchers to identify groups of people who are at the most risk for certain types of injuries, as well as to identify risk factors for these injuries. These analyses can also help to identify trends in the number of injuries at a specific time and place, which can lead to a better understanding of the causes of injuries and help to develop prevention strategies [25,26,27]. These analyses can also be used to create diagnostic criteria for different types of injuries and increase the effectiveness of health care in this area [28,29,30].

In our research, we aimed to examine the change in trauma volume, age distribution, mortality, the most frequently injured body regions, and injury mechanism during the first lockdown period during the COVID-19 pandemic in 2020, compared with the same period in 2019 in Lublin Province, Poland. We hypothesize that the trauma volume decreased with different trends in morbidity and injury mechanisms during the first national lockdown.

## 2. Materials and Methods

### 2.1. Background

Lublin Province is a voivodeship in south-eastern Poland with 2.09 million inhabitants (Figure 1) [31].

The presented paper is a retrospective cohort study, based on data derived from the administrative registry kept by the National Institute of Public Health’s National Institute of Hygiene (NIPH-NIH) within the framework of the National General Hospital Morbidity Study (NGHMS) [33]. NGHMS is the main source of hospitalized morbidity data in Poland. Each hospital in Poland is obligated to prepare a discharge report for inpatients and to deliver this data to the NIPH-NIH. Every single hospitalization is recorded in the database and described by the International Classification of Diseases Revision 10 (ICD-10) for the main diagnosis. Patients presenting trauma are coded according to codes included in Chapter XIX of the ICD-10.

Our study compared data from the 11 weeks of the first lockdown in 2020 (12 March–30 May) with the same period in 2019. The following variables were taken into account: patients’ ICD-10 code for the main diagnosis, patients’ ICD-10 code for the injury mechanism, patients’ age and gender, length of hospital stay, code of hospital ward, date of admission, and hospitalization outcome (death or survival).

We performed an analysis of all trauma admissions to 35 hospitals in Lublin Province, recorded in NGHMS, irrespective of referral type.

The inclusion criteria comprised having been diagnosed with an injury defined by an ICD-10 diagnostic code from S00 to T98.3 during a hospital stay between 12 March and 30 May in 2019 and 2020. We identified 12,129 records in 2019 and 5968 in 2020. The exclusion criteria were: patients with records with missing data, such as the ICD-10 code of the disease, age, gender, or date of admission. As a result of missing data, 1298 records in 2019 and 748 records in 2020 were excluded from the analysis. Iatrogenic injuries defined as ICD-10 code Y40-Y84 for the injury mechanism or ICD-10 code T80-T88 for the injury type were also excluded. Finally, 10,806 patients in 2019 and 5212 in 2020 met the inclusion criteria and did not meet the exclusion criteria. Injuries were classified by body region with the ICD-10 according to Table 1.

The mechanisms of injury were extracted from a database and classified with ICD-10 classification, with the range of V01-Y87, according to Table 2.

The primary analysis aimed to compare the mortality rate of hospitalized patients, as well as the frequency, category, mechanism, and area of injuries. Additionally, the patients’ average length of stay in the hospital and age were compared.

### 2.2. Statistical Analyses

All statistical analyses were conducted using IBM SPSS Statistics (Version 28) and R language (Version 4.1).

For continuous variables, the Mann–Whitney U test was used, with medians and interquartile ranges reported as descriptive statistics, along with the strength of the effect—r. Categorical variables were compared using the chi-square test, and subgroup analyses were performed with a Bonferroni correction. A correction for Yates’ continuity was applied to the 2 × 2 contingency tables. For contingency tables where more than 20% of cells had expected cell counts of less than 5, Fisher’s exact test was performed. Standardized residuals were calculated to assess the significance of differences between groups. These residuals measure the difference between the observed and expected frequencies in contingency tables, adjusted for sample size and distribution. An absolute value above 2 for a standardized residual indicates a significant difference between groups. This approach has been used in previous studies to identify important associations and differences [34]. In addition, Cramer’s V statistic was calculated, which is considered the strength of the effect. The use of standardized residuals and Cramer’s V statistic are common approaches in statistical analyses to identify significant differences and associations between variables [34]. In order to accurately compare events between years, relative risk ratios and confidence intervals for each were compiled.

The statistical significance level was set at *p* = 0.05.

## 3. Results

The sample consisted of a total of 16,018 records, which included 10,806 patients in 2019 and 5212 patients in 2020. Of the patients, 9837 were men (61%), and 6181 were women (39%). Specifically, in 2019, there were 6650 men (62%) and 4156 women (38%), while in 2020, there were 3187 men (61%) and 2025 women (39%).

Table 3 shows comparisons of the total hospital admissions by age range in 2019–2020. It also compares total admissions by age range, in-hospital mortality (death), mortality by sex, and ranged age in relation to the total number of admissions. The percentages represent the share of each event inside the columns.

A significant difference was observed between 2019 (67.5%) and 2020 (32.5%) in total admissions (RR = 2.07, 95% CI: 2.02, 2.13). In 2020, the proportion of admitted patients dropped from 16.3% to 10.4% in the age group of 1–17 (RR = 1.57, 95% CI: 1.43, 1.72) and from 40.4% to 39.5% in the 18–45 range (RR = 1.02, 95% CI: 0.98, 1.07), compared to the year 2019. In comparison, in 2020, there was an increase in patient admissions from 23.8% to 25.7% in the 46–65 range (RR = 0.93, 95% CI: 0.88, 0.98) and from 19.5% to 24.4% in the 65+ group (RR = 0.80, 95% CI: 0.75, 0.85). Total mortality increased from 0.77% in the year 2019 to 1.67% in 2020 (RR = 0.46, 95% CI: 0.34, 0.62). Furthermore, there were significant increases in mortality between 2019 and 2020 in the age ranges of 1–17 (from 0% to 0.56%) and in the 65+ group (from 2.67% to 4.80%, RR = 0.56, 95% CI: 0.39, 0.79). The significance of these differences was assessed using a chi-square test with a Bonferroni correction for subgroup analyses. The remaining comparisons were found to be statistically insignificant.

There was a significant difference in age between the groups (Table 4). Patients in 2020 were on average older (*Mdn* = 46, *IQR* = 35) than patients in 2019 (*Mdn* = 40, *IQR* = 39).

Table 5 contains frequency comparisons between 2019 and 2020 for the mechanisms of injury with which patients were hospitalized (according to the ICD-10).

The chi-square test was found to be statistically insignificant, χ^2^(14) = 20.61, *p* = 0.112, *V*_c_ = 0.06. Additional comparisons with a Bonferroni correction and standardized residuals also indicated no significant differences between groups.

The absolute number of injuries per body region decreased over the time of the lockdown period. Figure 2 shows a graphical presentation of changes in the number of injuries in the two analyzed periods.

Table 6 compares the frequencies between 2019 and 2020 for injuries with which patients were hospitalized by region of the body (according to the ICD-10).

Between 2019 and 2020, there was an increase in the percentages of injuries in body regions such as the upper limbs (from 28.9% to 33.2%, RR = 0.87, 95% CI: 0.83, 0.92) and lower limbs (from 29.3% to 32.2%, RR = 0.91, 95% CI: 0.87, 0.96). In contrast, there has been a decrease in the number of cases of head injuries (from 22.2% to 18.2%, RR = 1.22, 95% CI: 1.14, 1.31) and other injuries (from 7.1% to 3.2%, RR = 2.22, 95% CI: 1.86, 2.64).

The remaining comparisons did not reach statistical significance.

A graphical analysis of the absolute number of the most common diagnostic codes (according to the ICD-10) in the analyzed periods is presented in Figure 3.

A comparison of the frequency of the 20 most common diagnoses made at hospital admission (according to the ICD-10) in 2019–2020 is presented in Table 7. Patients could have only one primary diagnosis. The relative risk ratios in bold indicate significant differences between the groups.

Between 2019 and 2020, there was a decrease in the number of diagnoses such as superficial head injuries (from 6.62% to 1.06%, RR = 6.25, 95% CI: 3.78, 10.35), a foreign body in the cornea (from 6.32% to 0.46%, RR = 13.66, 95% CI: 6.45, 28.93), and sequelae of other fractures of the lower limbs (from 5.65% to 1.92%, RR = 2.94, 95% CI: 2.00, 4.33).

On the other hand, there was an increase in the percentage of open wounds to the scalp (from 5.76% to 8.27%, RR = 0.70, 95% CI: 0.56, 0.87), dislocation of the shoulder joint (from 2.56% to 4.17%, RR = 0.61, 95% CI: 0.45, 0.84), fracture of the lower end of the radius (from 8.24% to 10.85%, RR = 0.76, 95% CI: 0.63, 0.91), fracture of the lower end of both the ulna and radius (from 2.47% to 3.90%, RR = 0.63, 95% CI: 0.46, 0.88), open wounds to other parts of the wrist and hand (from 2.85% to 4.23%, RR = 0.67, 95% CI: 0.49, 0.92), fracture of the femoral neck (from 6.59% to 10.85%, RR = 0.61, 95% CI: 0.50, 0.74), and pertrochanteric fracture (from 5.26% to 9.06%, RR = 0.58, 95% CI: 0.47, 0.72), as shown in Table 2. The remaining comparisons did not reach statistical significance.

## 4. Discussion

This analysis confirms that the restrictions imposed by authorities during the first COVID-19-related lockdown influenced changes in trauma epidemiology in Lublin Province, Poland. We found a significant reduction in trauma admissions, with a decreased proportion of head injuries and an increased proportion of limb injuries. We did not observe changes in the most common trauma mechanisms. A higher mortality and a higher median age among trauma encounters were noticed in this study.

### 4.1. Trauma Volume

We found a total reduction in trauma cases during the first lockdown period in Lublin Province in Poland of almost 52% in 2020, compared with the same period in 2019 (*p* < 0.01). The decrease in trauma admissions was also observed in other countries [35,36]. Kreis et al. reported a 30% reduction in all emergency department visits and a 20% reduction in the absolute number of surgeries during the first lockdown period in a Level 1 trauma center in Germany, irrespective of injury severity [37]. However, Harris et al. noticed only a 17% decline in all severe trauma presentations in a major trauma center in Adelaide, Australia, which was substantially lower than in other regions. They concluded that it was caused by the less significant social restrictions enforced by the Australian government than in other countries [38]. Comparable results were presented by Bäckström et al. in the study of trauma admissions in Sweden, recorded in the Swedish trauma registry [39]. They noticed a reduction of 22% in all severe injuries. According to the authors, the smaller decline in trauma admissions, compared with other countries, was a result of the relatively lenient measures against COVID-19 in Sweden. In the multicenter analysis of trauma care during the lockdown period in the Northern Netherlands, a decrease of 37% in all severe trauma patients was recorded by the Emergency Department. A large, multicenter study of 12 395 patients from 85 trauma centers in the USA reported a 35% absolute decrease in trauma volume in April 2020 compared with April 2019 [40]. All injuries were included, and the majority of them were defined as minor or moderate severity. Another observational paper presented by Moyer et al., based on the trauma registry, reported a significant decrease (43%) in major trauma admissions during the COVID-19 era (*n* = 361) in France than in the same period the year before (*n* = 628) [41]. The total reduction in trauma admissions in our paper was one of the highest compared to the other countries. In our opinion, the severe restrictions implemented by government, to reduce the spread of COVID-19, significantly changed the mobility and everyday activity among the polish society, which decreased the risk of injuries. Furthermore, organizational problems in healthcare made access to medical services more difficult in Poland. Patients with minor injuries were likely to receive assistance in a general practitioner’s office or via telemedicine. We also observed admission differences among various age groups. There was a significant reduction in admissions among injured patients under 18, but in other age groups, changes did not reach a significant level. A recently published paper by Ohm et al. reported a similar observation that the reduction in admissions among the youngest part of society was higher than in other age groups [42]. Several other studies reported a reduction in injuries among children and adolescents [43,44,45]. A possible explanation for this observation is that the restrictions implemented by authorities mostly affected everyday activities among children and adults. Remote classes, postponed extracurricular activities, and spending more time at home decreased their risk of injuries.

### 4.2. Median Age

We noticed that the median age of injured patients significantly increased in the time of the first lockdown period from 40 years in the pre-COVID-19 era to 46 years in the first lockdown period. In a recently published paper by Adiamah et al., the same difference in two age groups was observed among major trauma patients [46]. What is important for the analysis of this study is a study conducted in one trauma center in Nottingham, United Kingdom (UK). Badach et al. reported a trend toward elderly patients but without statistical significance in a multicenter study in New Jersey trauma centers [7]. An increase in mean age was observed in a study of major trauma admissions in four hospitals in Finland presented by Riuttanen et al. [47]. On the other hand, in a large database study of trauma admission during the COVID-19 pandemic in Japan, a significant difference in mean age was not reported [48]. This study analyzed only major trauma and assessed the influence of COVID-19 over a longer time period than in our study. The potential explanation for the increase in median age of patients suffering trauma in Lublin Province is a different age structure among trauma admissions in the time of the first lockdown period compared with the previous year. Severe restrictions imposed by the Polish government, such as the stay-at-home policy, highly affected the everyday life of the youngest part of the Lublin Province residents (i.e., closing the schools, decrease in physical activity), which is reflected in the reduction in admissions among kids and young adults. COVID-19 restrictions did not affect the habits of elderly people to the same extent as in the case of other age groups, so restrictions did not reduce the risk of potential injury among them. Furthermore, it is well-known that elderly patients are usually the victims of low-energy injuries occurring at home. A stable amount of injuries among patients older than 65 with a reduction in admissions among the youngest part led to increase in median age.

### 4.3. Trauma Mortality

A significant increase in mortality was observed during the lockdown period in the presented paper. Importantly, the absolute number of deaths was even higher in 2020 (*n* = 87) than in 2019 (*n* = 83). A significant increase in mortality was observed among children and the elderly. Our findings are in line with previously published studies. Higher mortality was reported during COVID-19 restrictions among patients in a major trauma center in the UK, where an increase from almost 5% (2019) to close to 10% (2020) was reported. Injury severity and frailty were significant factors influencing mortality in this paper [46]. Nia et al. reported a significant increase in mortality (4% vs. 14%) over the time of a lockdown in a Level 1 trauma center in Austria, among all trauma admissions. Additionally, an analysis of trauma epidemiology during the first 2 months of COVID-19 restrictions in 2020 in three trauma centers in the Netherlands revealed an increase in 30-day mortality (1.0% vs. 2.3%) compared to the same period in 2019 [49]. The discussed paper covered a full range of injury severity, with the majority of participants suffering minor or moderate trauma. Contrary to the results presented in this study, we expected that trauma morality in the time of the first lockdown period was lower than in previous year. In our opinion, the relatively stable absolute number of deaths in the two analyzed periods reflects that there was no decrease in severe and potentially life-threatening injuries in the first wave of COVID-19 in Poland. In the light of that, in similar situations in the future, trauma services for the most severe trauma victims should be maintained.

### 4.4. Injury Mechanisms

The presented research did not provide descriptions of significant changes in injury mechanisms during the first lockdown period compared with the same interval in 2019. However, the ICD-10 code of the injury mechanism was recorded in only 42% of cases. Non-transportation injuries were much more common than transportation-related injuries in both study periods. Falls from a standing height, falls from a height, and being struck by an object were the three main causes of injury in the analyzed periods of 2019 and 2020. Car accidents were the main cause of injury in the group of transportation injuries. Chiba et al. reported a significant reduction in transportation injuries such as car versus pedestrian or motorcycle accidents and an increase in falls from a standing height in admissions to a large, Level 1 trauma center in Los Angeles [50]. According to the authors, the higher incidence of ground-level falls was influenced by stay-at-home orders. The incidence of car accidents was constant and, according to them, may have resulted from an increased level of patients transferred from other facilities. In the study by Ghafil et al., trends in trauma admissions (14,000 admissions) during the COVID-19 pandemic in 15 trauma centers in Los Angeles Country were analyzed, and they found no significant differences in transportation injuries and falls compared with the same period in 2020 [51]. Ojiima et al. assessed the impact of the COVID-19 pandemic on major trauma in Japan, and a significant decrease in motor-vehicle collision (MVC) injuries was reported during the lockdown period [48]. Moyer also reported a statistically significant reduction in cases only in MVCs among major trauma patients in France [41]. The stable number of transportation-related injuries observed in this study could be explained by the low level of road safety in Poland. According to the European Commission report, with 66 road fatalities per million inhabitants, Poland remains one of the most dangerous countries to drive in Europe. In Germany, researchers reported a 3.7% decrease in car accidents and a 3.2% increase in bike accidents in a group of more than 40,000 severely injured patients from the German trauma registry during the lockdown [52]. According to the authors, social restrictions and an increase in those who switched to working from home resulted in less traffic and a decrease in MVCs. The closure of sports facilities and events led to an increase in bike-related activities which, according to Colcuc et al., could be explained by increased bicycle sales in the first half of 2020. In Poland, 68% of the society maintained their physical activity during the lockdown period in Poland, but mostly at home [53]. The stable amount of bike injuries in this study was influenced by the severe restrictions imposed by the Polish government and the general attitude among Polish people to perform physical activity at home during the first lockdown period.

### 4.5. Anatomic Description of Injury

We observed a significant reduction in patient referrals with head trauma. Colcuc et al. reported contradictory results in their multicenter analysis and recognized an increase in the percentage of head injuries during the lockdown period [52]. According to them, the lower physical activity in society with increased interest in bike-related sports were the reasons for this phenomenon. We briefly discussed attitudes to physical activity among Polish people above, and in our opinion, limited outdoor activity slightly influenced the lower incidence of traumatic brain injury (TBI) in this study. The main reason for the opposite results of TBI admissions in our research was the different inclusion criteria. Colcuc et al. included only severely injured patients, while in this paper, we analyzed the entire trauma population of Lublin Province. Miekisiak et al., in the national analysis of TBI incidence in Poland, found that the volume of TBI in 2020 dropped by almost 25% in comparison with the previous six years, with a significant reduction in conservatively treated patients [54]. In several single-center analyses of TBI epidemiology during the lockdown period, a decrease in head trauma was reported during the time of COVID-19 restrictions [55,56,57,58,59]. In this paper, we found a significant decrease in TBI admissions during the lockdown period in the general population. A possible explanation includes lesser everyday activity during the lockdown and avoiding visiting the hospital with mild TBI. This is reflected in the reduction in referrals with a superficial injury of the head using the ICD-10 code found in this study.

An increase in the proportion of lower and upper extremity injuries in the first lockdown period was observed in this analysis. Our findings are in line with a report provided by Dolci et al. [60]. In the analysis of musculoskeletal trauma in the Cagliari Metropolitan Area (Italy), they found a reduction in the absolute number of extremity injuries with an increased proportion of lower and upper limb fractures. An analysis of a specific diagnosis revealed an increased proportion of distal radius fractures and proximal femur fractures, as in the results obtained in this study [60]. On the other hand, an evaluation of the eTrauma management platform (Open Medical, London, UK) performed by Sephton et al. showed no significant differences in the proportion of upper and lower limb trauma among orthopedic referrals [10]. Similar findings were reported in a study by Malige et al. [61]. An analysis of differences in orthopedic trauma in Pennsylvania during the COVID-19 outbreak, using the Pennsylvania Trauma System Foundation Registry, found no meaningful differences in the distribution of injuries [61]. The results obtained in studies regarding changes in the occurrence of orthopedic trauma are inconclusive. In our opinion, differences in the distribution of these injuries were highly influenced by government restrictions during the pandemic. The severe restrictions imposed by the government forced the Polish population to spend more time at home, which increased the risk of low-energy extremity injuries. The higher proportion of distal forearm fractures and proximal femur fractures found in this paper during the time of the first lockdown confirms that the higher occurrence of low-energy trauma remains a concern in this unique period. In the review article by Kim et al. regarding orthopedic trauma admission, an increase in the proportion of femur fractures in the time of the lockdown period was reported [62]. Recently published papers noticed that at the time of the COVID-19 outbreak, proximal femur fractures remained a serious health issue. In the multicenter analysis by MacDonald et al., a reduction in the total amount of orthopedic admissions was observed with a reduced absolute number and a higher proportion of proximal femur fractures [63]. On the other hand, in the study by Nunez, the absolute number of osteoporotic hip fractures in the first lockdown period remained constant [64]. In our opinion, a possible explanation for the patterns of hip fractures observed in our study is that these kinds of injuries commonly occur among the elderly at home. Furthermore, the restrictions imposed by authorities may be expected to reduce more of the trauma sustained outside than in households. A reduction in injuries among young adults and children, with a relatively lower decrease in the absolute number of femur fractures, explains the various proportions of these injuries in the two analyzed periods in Lublin Province.

This study has several limitations. First, this is a retrospective analysis. Second, the reports delivered by each hospital are often incomplete and filled out inappropriately, and this could hamper the analysis. Third, the NIPH registry is an administrative database and does not meet the criteria of a trauma registry. Other factors, such as injury severity, cannot be assessed because there is no high-quality trauma registry in Poland. On the other side, the analysis was carried out on the biggest and most well-known database in Poland including a high volume of records which allows for more reliable conclusions than studies on smaller populations.

## 5. Conclusions

Significant reductions in trauma volume and specific types of injuries were observed during the lockdown period, reflecting a shift in the pattern of trauma cases presented at healthcare facilities in Lublin Province corresponding to the worldwide trend. A reduction in the most life-threatening head and neck injuries was noticed. The proportion of proximal femur fractures and distal forearm fractures relatively increased over the time of the COVID-19 outbreak. There were no changes observed in the most common injury mechanisms and gender distribution in the two analyzed periods. An increase in mortality among children and the elderly was found, and the median age of injured patients was higher in the COVID-19 era. This shift has implications for the future preparedness of healthcare services during pandemics or other global disasters. The presented findings recommend adaptive strategies in healthcare systems to manage the evolving nature of trauma cases in the face of future disasters and changes in societal behavior.

## Figures and Tables

**Figure 1 healthcare-11-02436-f001:**
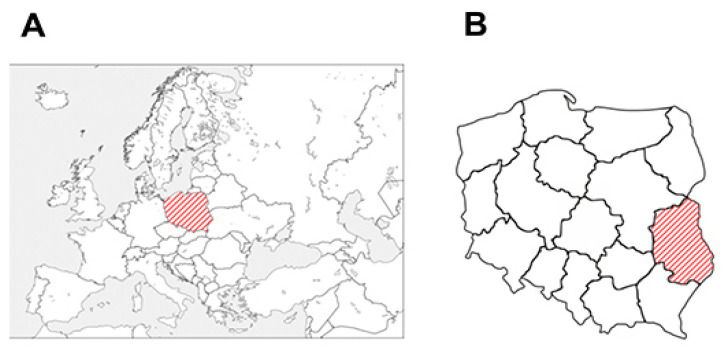
Approximate locations of study area, Lublin Province, Poland [32]. (**A**) location of Poland; (**B**) Lublin Province location.

**Figure 2 healthcare-11-02436-f002:**
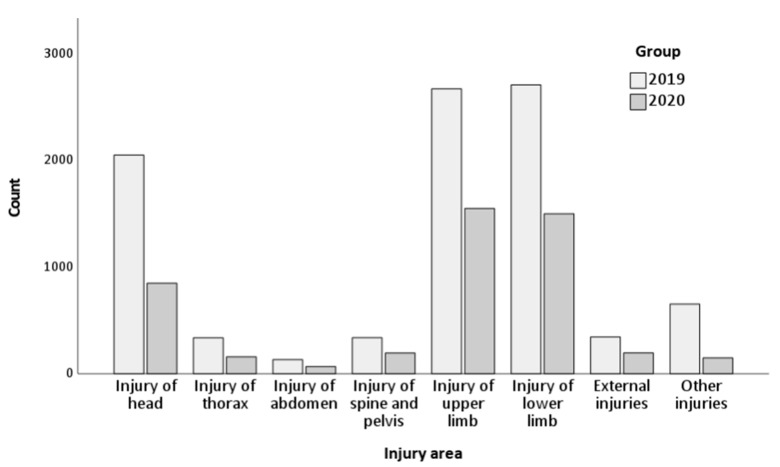
Diagram of an absolute number of injuries in the various body area.

**Figure 3 healthcare-11-02436-f003:**
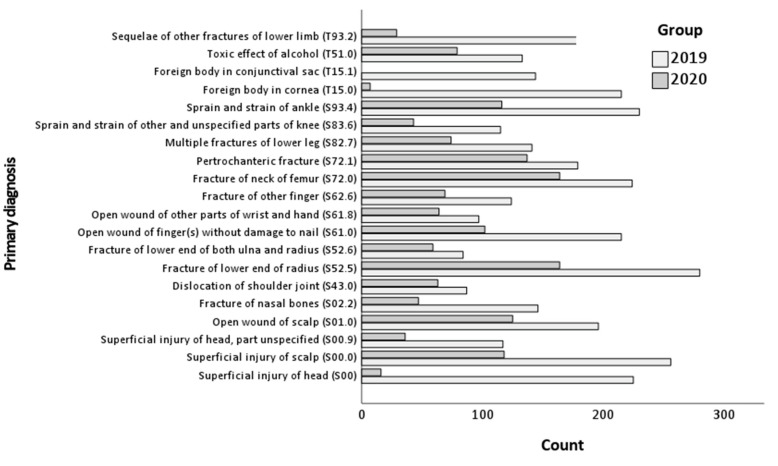
Diagram of the 20 most common diagnostic codes 2019–2020.

**Table 1 healthcare-11-02436-t001:** Classification of injury by body region according to the ICD-10.

Area of Injury	ICD-10 Code
Injury of head	S00–S11.9, S15–S19.9, T06.0–T06.1, T90.0–T91.9
Injury of thorax	S20–S21.9, S22.2–S22.9, S23.2–S23.5, S24.3–S29.9
Injury of abdomen	S30–S31.8, S34.4–S39.9
Injury of spine and pelvis	S12–S14.6, S22.0–S22.1, S23.0–S23.1, S24.0–S24.2, S32.0–S34.3, T08, T09.3, T09.4
Injury of upper limbs	S40–S69.9, T05.0–T05.2, T10–T11.9, T92.0–T92.9
Injury of lower limbs	S70–S99.9, T05.3–T05.5, T12–T13.9, T93.0–T93.9
External injuries	T00–T01.9, T14.0, T20–T35.7, T66–T68, T95.0–T98.3
Others	S00.0–T87 not included above

**Table 2 healthcare-11-02436-t002:** Classification of injury mechanism according to the ICD-10.

Injury Mechanism	ICD-10 Code
Communication injuries	
Pedestrian injured in collision	V01–V09
Pedal cyclist injured in collision	V10–V19
Motorcycle rider injured in collision	V20–V29
Car occupant injured in collision with pedestrian or animal	V40–V49
Occupant of pick-up truck or van injured in collision with pedestrian or animal	V50–V79
Other communication injuries.	V30–V39, V80–V99, Y85
Non-communication injuries	
Falls from same level	W00–W09, W18
Fall on and from height	W10–W17, Y30
Assaulted by another person	W32–W34, W50–W52, X88–Y05, Y07–Y09, Y20, Y22–Y23
Intentional self-harm	X70–X84, Y87
Attacked by an animal	W53–W59, X20–X29
Contact with lifting and transmission devices	W24, W28–W31
Struck by thrown, projected, or falling object	W20–W23, W25–W27, W44–W45, W60, Y28–Y29
Exposure to high or low temperature, high or low pressure, electric current, forces of nature	W35–W41, W85–W92, X00–X19, X30–X33, Y25–Y27
Other causes	V00–Y87 not included above

**Table 3 healthcare-11-02436-t003:** Comparison of total hospital admissions and patient mortality in 2019–2020.

Variable	2019		2020		*p*	RR	95% CI	
	*n*	%	*n*	%			*LL*	*UL*
Total admissions	10,806	67.5	5212	32.5	<0.001	2.07	2.02	2.13
Admissions by age								
1–17	1754	16.3	540	10.4	<0.001	1.57	1.43	1.72
18–45	4354	40.4	2056	39.5		1.02	0.98	1.07
46–65	3157	23.8	1339	25.7		0.93	0.88	0.98
65+	1513	19.5	1271	24.4		0.80	0.75	0.85
Mortality	83	0.77	87	1.67	<0.001	0.46	0.34	0.62
Mortality by sex								
Male	52	0.78	52	1.63	<0.001	0.48	0.33	0.70
Female	31	0.75	35	1.73	<0.001	0.43	0.27	0.70
Mortality by age								
1–17	0	0	3	0.56	0.013 ^a^	n/a	n/a	n/a
18–45	6	0.14	5	0.24	0.346 ^a^	0.57	0.17	1.85
46–65	21	0.82	17	1.27	0.173 ^b^	0.64	0.34	1.22
65+	56	2.67	61	4.80	0.001 ^b^	0.56	0.39	0.79

Note. ^a^ Fisher’s exact test, ^b^ continuity correction, RR—risk ratio, CI*—*confidence interval; *LL—*lower limit; *UL*—upper limit.

**Table 4 healthcare-11-02436-t004:** Comparison of patients’ age 2019–2020.

Variable	2019 (*n* = 10,806)		2020 (*n* = 5212)		*Z*	*p*	*r*
	*Mdn*	*IQR*	*Mdn*	*IQR*			
Age of patients	40	39	46	35	−11.44	<0.001	0.11

Note. *Mdn*—median, *IQR*—interquartile range, Z—Z test statistic, *r*—strength of effect.

**Table 5 healthcare-11-02436-t005:** Comparison of mechanisms of patient injuries in 2019–2020.

Variable	2019		2020		RR	95% CI	
	*n*	%	*n*	%		*LL*	*UL*
Mechanism of injury							
Pedestrian injured in collision	52	2.3	24	2.2	1.24	0.76	2.00
Pedal cyclist injured in collision	96	1.1	54	1.0	1.02	0.73	1.41
Motorcycle rider injured in collision	46	2.5	25	2.0	1.05	0.65	1.71
Car occupant injured in collision with pedestrian or animal	105	0.2	49	0.2	1.22	0.88	1.71
Occupant of pick-up truck or van injured in collision with pedestrian or animal	7	0.0	5	0.0	0.80	0.25	2.52
Other communication injuries.	2	39.5	1	37.0	1.14	0.10	12.59
Fall on same level	1674	6.2	895	7.0	1.07	1.00	1.14
Fall on and from height	264	2.4	169	2.0	0.89	0.74	1.07
Assaulted by another person	101	0.2	49	0.3	1.18	0.84	1.65
Intentional self-harm	7	1.6	7	1.0	0.57	0.20	1.63
Attacked by an animal	66	3.0	25	3.1	1.51	0.95	2.38
Contact with lifting and transmission devices	126	11.1	76	10.4	0.95	0.72	1.25
Struck by thrown, projected, or falling object	471	1.1	251	1.5	1.07	0.93	1.24
Exposure to high or low temperature, high or low pressure, electric current, forces of nature	47	27.7	37	31.1	0.73	0.47	1.11
Other causes	1172	1.2	752	1.0	0.89	0.82	0.96
Mechanism of injury—general category							
Communication injuries	308	10.1	158	9.5	1.06	0.88	1.27
Non-communication injuries	2756	89.9	1509	90.5	0.99	0.97	1.01

Note. RR—risk ratio, CI*—*confidence interval; *LL—*lower limit; *UL*—upper limit.

**Table 6 healthcare-11-02436-t006:** Comparison of the frequency of injuries (by area) suffered by patients in 2019–2020.

Variable	2019		2020		RR	95% CI	
	*n*	%	*n*	%		*LL*	*UL*
Area of injury							
Injury of head	2048	22.2	847	18.2	1.22	1.14	1.31
Injury of thorax	336	3.6	158	3.4	1.07	0.89	1.29
Injury of abdomen	132	1.4	67	1.4	0.99	0.74	1.33
Injury of spine and pelvis	337	3.7	194	4.2	0.88	0.74	1.04
Injury of upper limb	2669	28.9	1547	33.2	0.87	0.83	0.92
Injury of lower limb	2706	29.3	1498	32.2	0.91	0.87	0.96
External injuries	344	3.7	195	4.2	0.89	0.75	1.06
Other injuries	651	7.1	148	3.2	2.22	1.86	2.64

Note. RR—risk ratio, CI*—*confidence interval; *LL—*lower limit; *UL*—upper limit.

**Table 7 healthcare-11-02436-t007:** Comparison of the 20 most common diagnoses given to patients in 2019–2020.

Variable	2019		2020		RR	95% CI	
	*n*	%	*n*	%		*LL*	*UL*
Primary diagnosis							
Superficial injury of head	225	6.62	16	1.06	6.25	3.78	10.35
Superficial injury of scalp	256	7.53	118	7.80	0.96	0.78	1.19
Superficial injury of head, part unspecified	117	3.44	36	2.38	1.45	1.00	2.09
Open wound of scalp	196	5.76	125	8.27	0.70	0.56	0.87
Fracture of nasal bones	146	4.29	47	3.11	1.38	1.00	1.91
Dislocation of shoulder joint	87	2.56	63	4.17	0.61	0.45	0.84
Fracture of lower end of radius	280	8.24	164	10.85	0.76	0.63	0.91
Fracture of lower end of both ulna and radius	84	2.47	59	3.90	0.63	0.46	0.88
Open wound of finger(s) w/o damage to nail	215	6.32	102	6.75	0.94	0.75	1.18
Open wound of other parts of wrist and hand	97	2.85	64	4.23	0.67	0.49	0.92
Fracture of other finger	124	3.65	69	4.56	0.80	0.60	1.07
Fracture of neck of femur	224	6.59	164	10.85	0.61	0.50	0.74
Pertrochanteric fracture	179	5.26	137	9.06	0.58	0.47	0.72
Multiple fractures of lower leg	141	4.15	74	4.89	0.85	0.64	1.12
Sprain and strain of other and unspec. Parts of knee	115	3.38	43	2.84	1.19	0.84	1.68
Sprain and strain of ankle	230	6.76	116	7.67	0.88	0.71	1.09
Foreign body in cornea	215	6.32	7	0.46	13.66	6.45	28.93
Foreign body in conjunctival sac	144	4.24	0	0.00	n/a	n/a	n/a
Toxic effect of alcohol	133	3.91	79	5.22	0.75	0.57	0.98
Sequelae of other fractures of lower limb	192	5.65	29	1.92	2.94	2.00	4.33

Note. RR—risk ratio, CI*—*confidence interval; *LL—*lower limit; *UL*—upper limit.

## Data Availability

The data presented in this study are available on request from the corresponding authors.

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
