# Peer review of "Evaluating Changes in Trauma Epidemiology during the COVID-19 Lockdown: Insights and Implications for Public Health and Disaster Preparedness"

_healthcare, 2023, doi:10.3390/healthcare11172436_

Round 1

Reviewer 1 Report

The manuscript presented for evaluation discusses the issue concerning the influence of covid-19 lockdown on the epidemiology of trauma in a group of patient of 35 hospitals located in Polish district: Lublin. The issue is of importance and vital from the point of view of sustainability of healthcare to prepare for the future lockdown caused by various health hazards. Nevertheless, I would suggest few revisions to improve the research.

The introduction

The problem of lockdown introduced in Poland and the ability to succumb to the new rules is described concerning the Polish and the European trend.

I would add to the description how the health system was modified adequately to needs and the one- name hospitals were introduced limiting the preparedness for trauma and how many of those were created in Lublin province during the 2020. The aim of the new types of hospitals. Also, how the ICD classification is present in everyday diagnosis of patients.

Material and methods

There is a clear description of the place where the research took place. However, to make it clearer maybe a visuals with a map, showing location of Poland and the district itself would be helpful to place it.

Also, I would distinguish a part describing “statistical analyses” to make it easier to find.

The authors state:

This approach has been used 143 in previous studies to identify important associations and differences.”.  Please provide references concerning that statement.

Next, line 154 the authors state:

“The use 145 of standardised residuals and Cramer's V statistic are common approaches in statistical 146 analyses to identify significant differences and associations between variables. In order to 147 accurately compare events between years, relative risk ratios and confidence intervals for 148 them were compiled.” - provide a reference for the statement.

I would also move the last sentence from lines 150-151 as a first one, introducing to the analyses.

Results

The decryption is very informative; however, I would advise to make the percentages of participants as a whole number, not 61.4%, but 61%; not 38.6% but 39%. It is very odd to have a 0.4% of a participant. Furthermore, if there is 61% of men, 39% is left for the female population. The reader can conclude those second numbers.

Line 184 “The chi-square test was found to be statistically insignificant, χ2(14) = 20.61, p = .112, 184 Vc = .06.” the focus should be placed on significant differences. Those not significant are not worth to describe.

Discussion

The section is well built, divided into smaller sections, making it easier to follow.

The division of the sections semes to follow the importance of emergeing themes.

I would include two more recent studies in the discussion:

1.     Rajput, K., Sud, A., Rees, M. et al. Epidemiology of trauma presentations to a major trauma centre in the North West of England during the COVID-19 level 4 lockdown. Eur J Trauma Emerg Surg 47, 631–636 (2021). https://doi.org/10.1007/s00068-020-01507-w

2.     Pulak Vatsya, Siva Srivastava Garika, Samarth Mittal, Vivek Trikha, Vijay Sharma, Rajesh Malhotra, Lockdown imposition due to COVID-19 and its effect on orthopedic emergency department in level 1 trauma center in South Asia, Journal of Clinical Orthopaedics and Trauma, https://doi.org/10.1016/j.jcot.2022.101826.

Conclusion

There are based on the results provided a reflection concerning the changing landscape for the traumas and need to adjust healthcare solution towards those changes.

All in all, due to the importance of conducted research, I advise to accept the paper for publication in the Journal after introducing suggested minor revisions.

Author Response

Respected Reviewers,

Thank you very much for your review. We appreciate your suggestions and comments, which will undoubtedly improve the quality of our manuscript.

We wish to submit a revised manuscript, “Evaluating Changes in Trauma Epidemiology During the COVID-19 Lockdown: Insights and Implications for Public Health and Disaster Preparedness” for further consideration by Healthcare.

We confirm that we have considered all the valuable reviewer’ comments and applied changes to the manuscript. The improvements are marked in yellow in the manuscript.

Reviewer 1

Comments

Response

I would add to the description how the health system was modified adequately to needs and the one- name hospitals were introduced limiting the preparedness for trauma and how many of those were created in Lublin province during the 2020. The aim of the new types of hospitals. Also, how the ICD classification is present in everyday diagnosis of patients.

However, to make it clearer maybe visuals with a map, showing location of Poland and the district itself would be helpful to place it.

The points have been added to the manuscript.

The map has been added.

“This approach has been used 143 in previous studies to identify important associations and differences.”.  

The reference has been added.

“The use of standardised residuals and Cramer's V statistic are common approaches in statistical 146 analyses to identify significant differences and associations between variables. In order to accurately compare events between years, relative risk ratios and confidence intervals for them were compiled.” -

The reference has been added.

I would advise to make the percentages of participants as a whole number, not 61.4%, but 61%; not 38.6% but 39%

Where possible, the modification was applied, however when citing others results we have refrained from such change.

We have added proposed references.

Reviewer 2 Report

Abstract - The final statement lacks clarity - Our research indicates that the altered approach facilitated during such periods is essential for delivering tailored help to trauma patients. What is the altered approach that is required in the context of a reduction in the total number of patients? Line 25 - Specify country for international readers.

Introduction - The study is well-justified. Line 51 should be expanded to identify the key changes in the most common injury mechanism in mean age to give the reader literature context when reading this study's results. Line 70 - correct "what" to "which". Line 71 - clarify that the number of cases and deaths reported represented the total since the start of the pandemic (or start of aforementioned lockdown period).

Methods - Line 93 - explain what a voivodeship is for international readers. Line 112 - correct "exclusions" to "exclusion". Given the known limitations of retrospective cohort studies of this type, provide an indication of the number of patients excluded based on missing data. Table 2 - it appears "Fall on or from a height" is not complete (same for Table 5 in Results). Was data normality assessed prior to determining the use of Mann Whitney U test and presentation of median and IQR for each continuous variable? What is the justification for including p=0.05 in the statistical significance range, rather than the standard p<0.05? How were inter-hospital transfers handled, e.g., if patients were admitted to a smaller hospital but then required subsequent transfer and admission to a higher level facility? Were these potential duplicate entries identified and consolidated? Ethical statement required. Was this study reviewed by a Human Research Ethics Committee or was a waiver of consent approved?

Results - Ensure to include p values in the results descriptions, e.g., Line 145 for overall male/female difference. Ensure p values are provided after RR and 95% CIs. How is "the absolute number of injuries in per body regions decreased in 2020" different from total hospital admissions if only main diagnosis was considered? Were other diagnosis codes also looked at for each polytraumatized patient? Fig 1 - what were the findings of Chi-square tests, particularly for head, limb and other injuries which look substantially different between the 2 years?

Discussion - Line 220 - specify these findings were in 2020/lockdown. Line 264 - the importance described is not clear. Line 280 - provide a reference to support this statement. Line 305 - replace the term "prove" for a more appropriate scientific term, e.g., demonstrate/show. The subheadings used in the Discussion could be more descriptive of the actual findings, e.g. instead of just "Trauma mortality", "Trauma mortality was higher in COVID lockdown", etc. Other limitations to be discussed include the absence of prehospital data which likely underestimates total mortality given patients that expire prior to hospital admission are not included. Mortality was limited to in-hospital only and may have missed later mortality resulting from trauma complications (especially given the precedent to move patients out of hospital as quickly as possible during the pandemic). Patients not admitted to hospital are not captured so many minor injuries would not be captured.

Some minor spelling errors identified and listed above in General Comments for correction.

Author Response

Respected Reviewer,

Thank you very much for your review. We appreciate your suggestions and comments, which will undoubtedly improve the quality of our manuscript.

We wish to submit a revised manuscript, “Evaluating Changes in Trauma Epidemiology During the COVID-19 Lockdown: Insights and Implications for Public Health and Disaster Preparedness” for further consideration by Healthcare.

We do confirm that we have taken all the valuable reviewers’ comments under consideration and applied changes to the manuscript. The improvements are marked in yellow in the manuscript.

provide an indication of the number of patients excluded based on missing data.

Information about number of patients excluded based on missing data was incorporated.

Table 2 - it appears "Fall on or from a height" is not complete (same for Table 5 in Results).

Table 2 and 5 was corrected according to the reviewer suggestion

Was data normality assessed prior to determining the use of Mann Whitney U test and presentation of median and IQR for each continuous variable?

Yes, an assessment of data normality was conducted utilizing the Kolmogorov-Smirnov and Anderson-Darling tests, both of which revealed significant deviations from the assumption of normal distribution.

What is the justification for including p=0.05 in the statistical significance range, rather than the standard p<0.05?

A significance threshold is established at a p-value of p=0.05, indicating that outcomes with p-values below this level are considered statistically significant.

How were inter-hospital transfers handled, e.g., if patients were admitted to a smaller hospital but then required subsequent transfer and admission to a higher-level facility? Were these potential duplicate entries identified and consolidated? 

All the authors are unanimous in term of accepting above-mentioned changes. We hope that the version submitted will meet the criteria to be issued in Healthcare.

Thank you

Reviewer 3 Report

The work is of high quality and interest for the field of research. It has a very interesting sample that guarantees the representativeness, validity and reliability of the results obtained. For all these reasons, it is considered suitable for publication in the current magazine. As a suggestion for improvement, it is proposed for future studies to add data on work activity and mobility in the period analyzed. There is a clear relationship between the reduction in traffic and work activity during the pandemic and the accident rate in both sectors. It would be advisable to analyze these variables and relate said decrease with the data included in the present investigation.

Author Response

Respected Reviewers,

Thank you very much for your review. We appreciate your suggestions and comments, which will undoubtedly improve the quality of our manuscript.

We wish to submit a revised manuscript, “Evaluating Changes in Trauma Epidemiology During the COVID-19 Lockdown: Insights and Implications for Public Health and Disaster Preparedness” for further consideration by Healthcare.

Thank you for the suggestion, we will incorporate the data in our future research.  

All the authors are unanimous in term of accepting above-mentioned changes. We hope that the version submitted will meet the criteria to be issued in Healthcare.

Thank you

Katarzyna Naylor